# Experimental Study on the Carbon Sequestration Benefit in Urban Residential Green Space Based on Urban Ecological Carrying Capacity

**Yinglin Gong [1,2] and Xiaoyong Luo [1,*]**

1  School of Civil Engineering, Central South University, 22 Shaoshan Road, Changsha 410075, China; 70037@hutb.edu.cn
2  Design and Art Institute, Hunan University of Technology and Business, 569 Yuelu Avenue, Changsha 410205, China
*  Correspondence: csu-luoxy@csu.edu.cn

**Abstract:** The $CO_2$ concentration of urban residential green space in Changsha was experimentally investigated. Based on the experimental results, the variation characteristics and influencing factors of $CO_2$ concentration in residential areas were analyzed considering both the measuring time and ecological plant structure. Then, through the concept of urban ecological bearing capacity, the carbon sequestration index of the urban residential areas was proposed in this paper. Finally, the regulating effects of varieties of vegetation on $CO_2$ concentration among four urban residential areas were deeply analyzed and discussed. Results showed that green space with an ecological plant structure of trees-shrubs-grass exhibited greater improvement in the environmental carbon balance than those of shrubs or grass, and the atmospheric carbon sequestration capacity was significantly affected by the total quantity of the green space.

**Keywords:** $CO_2$ concentration; seasonal change; green space; influencing factors; bearing capacity

## 1. Introduction

The urban residential areas, as an essential component of ecosystem, have become a topic of everlasting interest to researchers and practitioners. In order to maintain the sustainable development of the urban residential areas, during the preceding decades, much effort has been directed towards the exploration on the ecological environmental bearing capacity of the green spaces [1,2], the coupling effects of human activities and variations of natural ecological environment [3], as well as the environmental amenity in urban residential areas [4,5]. Recently, due to the unceasing acceleration of global climate change (also known as the greenhouse effect) induced by the uncontrolled emission of $CO_2$ and other greenhouse gases [6], the concept of low-carbon development has been treated as the main theme on the topic of sustainable development in urban residential areas around world [7,8].

So far, in order to mitigate the greenhouse effect, a number of countries have introduced several limitation targets against carbon emissions to aim to achieve efficient carbon sequestration. For example, China has pledged to achieve the target of carbon neutrality by 2026. In detail, the main components of the greenhouse gases are $H_2O$, $CO_2$, $CH_4$, and CO. Specifically, the main route of the carbon cycle can be demonstrated as follows: $CO_2$ from atmosphere is firstly absorbed by land and marine plants, then returned to the atmosphere through biological and geological processes and human intervention. During this cycle, the effects of $CH_4$ and CO are relatively moderate when compared with $CO_2$ [9,10]. Due to the fact that $H_2O$ is usually not affected by human activities and CO is an indirect element, the $CO_2$ can be treated as the most contributing factor that enhances the greenhouse effect [10].

Therefore, a number of countries have conducted a series of programs to monitor the urban atmospheric $CO_2$ concentration [10–12].

At present, since most research regarding carbon sequestration mainly focuses on the bearing capacity of land, forests, lakes, and other large-scale spaces, the studies on smaller-scale green spaces are limited. For example, Rowntree and Nowak [13] estimated the carbon storage of urban forests in the United States, then analyzed the carbon absorption for urban green space and the reduction effect on carbon emission caused by human activities in two cities; Idso et al. [14,15] studied the distribution of near-surface atmospheric $CO_2$ concentration in Phoenix based on the inverse distance weight interpolation method and indicated that the concentration was significantly higher before sunrise than that of middle of day, and the causes were concluded to be the coupling effects of the atmospheric vertical mixing and vegetation photosynthesis; Wentz et al. [16] addressed the correlations among near-surface atmospheric $CO_2$ concentration, urbanization level (in terms of population, average traffic flow, and employment), and the vegetation coverage rate based on mobile monitoring data with regression analysis; Henninger et al. [17] studied the near-surface atmospheric $CO_2$ concentration in Essen based on mobile monitoring data, and the results showed that there were significant variations in different measuring areas and times for the concentration, and the average value in cities was 8.9% higher than that in suburbs. In addition, the research also found that the concentration variation was greater in winter when compared with summer, which resulted from the seasonal alterations in carbon emissions, the efficiency of the vegetation photosynthesis, and atmospheric conditions.

Concretely, with the continuous improvement of residents' living conditions, green plants have been treated as an indispensable component for the construction of modern urban residential areas [18–25]. For instance, Liang [21] studied the greenhouse effect by focusing on the green plants in residential areas and pointed out that the special physiological process of green plants, including carbon fixation and oxygen releasing, can contribute to the mitigation of the greenhouse effect. Meanwhile, in order to achieve eco-friendly and low-carbon development in both urban and rural areas, several countries have also issued a series of guiding documents specific to green plants [26–28].

According to previous studies, most researchers believed that the ecological environmental benefits resulting from the carbon sequestration of green plants were chiefly dependent on the efficient areas of leaves for completing the photosynthesis. Such benefits were usually determined by measuring the green quantity in certain regions [29–32], and plant carbon sequestration ability was generally calculated based on the relative biomass [33]. Hence, in this research, the $CO_2$ concentration is treated as the main operable factor for evaluating the effect of greenhouse gas on the urban ecological environmental condition. The relationship between the quality of the green space and the ecological environmental bearing capacity will be analyzed based on the measurement of $CO_2$ concentration in several sampling points.

Therefore, based on the above assertions, the main objective of the current research is to investigate the effects on the $CO_2$ concentration resulting from measuring time, location, and the plant community structure of the green spaces in urban residential areas. It is believed that the results can provide a theoretical basis and practice support for planning and constructing green spaces in order to efficiently improve the ecological environment in urban residential areas.

## 2. Materials and Methods

### 2.1. Observation Sites Setup

The observation sites were set up in four urban residential communities, which were located in the North (Xiangjiang River One and Huasheng New Band, aka XRO and HNB, respectively), central (Tongtai Meiling Court, aka TMC) and South (Ginkgo biloba home, aka GBH) areas of Changsha City in China, respectively. Specifically, in those communities, the plant spaces were all constructed after 2007, so the comparability for relevant analysis can be guaranteed. All sampling points were set up in accordance with

the principle of randomness and uniformity and arranged in intersection points with help of GPS positioning. Meantime, the locations of these sampling points were fine-tuned according to the plant community structure of the green spaces. The detailed information of the plant community structure is listed in Table 1. In addition, in order to obtain the spatial gradient of $CO_2$ concentration in the urban residential areas, for each observation site, the measuring areas were separated into central zone, transition zone, and edge zone.

**Table 1.** Plant community structure of green spaces and vegetation coverage.

| Green Space Type | Vegetation Coverage (%) | | |
|---|---|---|---|
| | Trees | Shrubs | Grass |
| Trees-Shrubs-Grass | 70 | 50 | 50 |
| Shrubs-Grass | 30 | 80 | 20 |
| Lawn | 30 | 40 | 60 |
| Trees | 100 | 0 | 0 |
| Shrubs | 0 | 100 | 0 |
| Grass | 0 | 0 | 100 |

Concretely, the ground green space chiefly consisted of shrubs, grass, and the trunks of trees. Therefore, the total green quality in this research is determined as the sum of the areas of the shrubs, grass, and the vertical projection of trees. Additionally, by considering the overlapping effect of the above-mentioned areas, the value of the vegetation coverage for the different varieties of vegetation can be eventually obtained.

Moreover, the bearing capacity of residential green space refers to the ability of the ecosystem to maintain its normal operation under various conditions. The relative bearing capacity index (also known as bearing capacity index ratio) is generally determined by the ratio of the measured value for ecosystem's bearing capacity and the critical value for that in certain conditions. Specifically, when the bearing capacity index ratio is larger than 1, it indicates that the pressure of the whole ecosystem in the region is greater than the standard, and the ecosystem can no longer bear economic activities. When the bearing capacity index equals 1, it indicates that the pressure generated by the residential economic activities of the residential green ecosystem is equal to the supporting force of the system, which is within the bearing capacity range. When the bearing capacity index is less than 1, it indicates that the pressure faced by the ecosystem does not exceed the maximum value it can bear, and the threat brought by the economic activities of residents in the community has not caused a relatively obvious impact, so the overall ecosystem tends to develop healthily. When the number of impact factors becomes lower, the ecosystem effect will be better, and people will feel more comfortable and work more efficiently.

After the geographical positions of the sampling points were determined, measuring instruments were placed in the center of each point. Additionally, the canopy density and plant community structure were measured. In this research, the average $CO_2$ concentrations from two comparison sample points were treated as the background data of the ecological environment for the urban residential areas. In detail, the comparison sampling points were located on the east side of GBH and HNB, with a small number of surrounding green landscape environments, respectively. The detail information for the sampling points is listed in Table 2.

**Table 2.** Basic information of sample points in residential green space.

| Sample Point | Location | Plant Community Type | Canopy Density |
|---|---|---|---|
| Comparison sample point 1 | 112°98′73″ E, 28°14′65″ N | - | - |
| Comparison sample point 2 | 112°98′30″ E, 28°21′70″ N | - | - |
| A | 112°96′59″ E, 28°29′96″ N | Trees-Shrubs-Grass | 0.75 |
| B | 112°96′55″ E, 28°29′95″ N | Shrubs-Grass | 0.45 |
| C | 112°96′55″ E,28°29′92″ N | Shrubs-Grass | 0.45 |
| D | 112°98′14″ E, 28°21′71″ N | Trees-Shrubs-Grass | 0.75 |
| E | 112°98′09″ E, 28°21′71″ N | Trees-Shrubs-Grass | 0.75 |
| F | 112°98′05″ E, 28°21′73″ N | Shrubs-Grass | 0.45 |
| G | 112°99′88″ E, 28°14′46″ N | Trees-Shrubs-Grass | 0.75 |
| H | 112°99′88″ E, 28°14′46″ N | Shrubs-Grass | 0.45 |
| I | 112°99′88″ E, 28°14′46″ N | Lawn | 0.30 |
| J | 112°98′82″ E, 28°14′57″ N | Lawn | 0.30 |
| K | 112°98′86″ E, 28°14′57″ N | Lawn | 0.30 |
| L | 112°98′89″ E, 28°14′57″ N | Lawn | 0.30 |

*2.2. Measurements Method*

In this research, the instrument for measuring the $CO_2$ concentration is Bohu intelligent $CO_2$ detector. It is based on gold-plated infrared photoconductivity technology and capable of automatic calibrating with a measurement accuracy of ±30 ppm. During the measuring process, the self-contained software of the instrument was used to record $CO_2$ concentration data per 60 s due to its sampling response time.

In order to guarantee the measured results are representative of spatio-temporal and ecological benefit factors, in this research, the measurements were conducted on a day when meteorological conditions and the weather conditions were relatively stable. During the measuring process, the instrument was kept within a certain distance from the ground (generally 1.4 m, and the monitoring time ranged from 8:00 to 18:00), as shown in Figure 1. To avoid the influences of atmospheric temperature and humidity, data collection was carried out within an interval of two hours with a detection time of 5 min.

Moreover, in order to obtain the annual variation of $CO_2$ concentration in urban residential green space, monthly measurements of $CO_2$ concentration were also conducted between the 20th and 25th from May 2019 to May 2020. During the data analysis process, the average value of measured $CO_2$ concentration of sample points was compared individually to indicate the environmental background data of the urban residential areas.

After obtaining the measuring data, the two-way and one-way variance analysis was conducted on the diurnal variation of measured $CO_2$ concentration based on difference method by considering the monthly variation and plant community structures, which was attempted to examine the influences resulting from both the measuring time and the location. In detail, relevant calculation methods are shown in Tables 3 and 4, respectively.

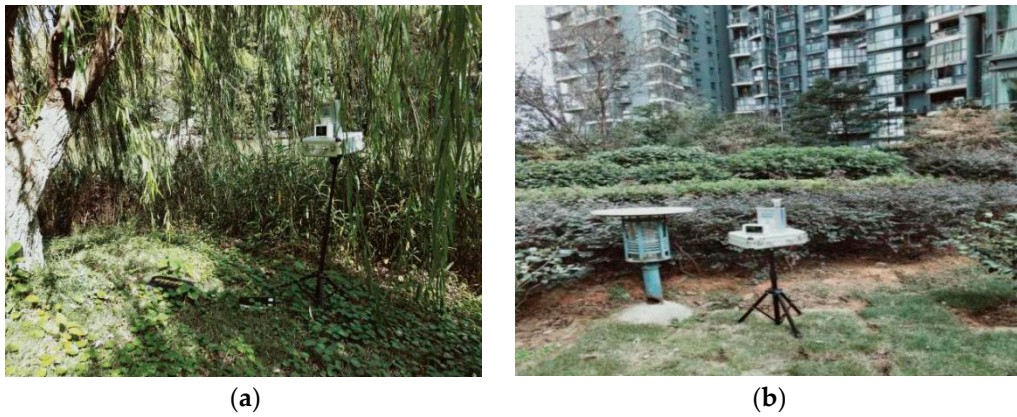

(a)            (b)

**Figure 1.** Photographs of observation sites: (**a**) XRO; (**b**) HNB.

**Table 3.** Variance analysis of regional samples and time $CO_2$ concentration of urban residential green space.

| | Quadratic Sum | Degree of Freedom | Mean Square | Mean Square Percentage | Probability ($\alpha$ = 0.05) |
|---|---|---|---|---|---|
| | **SS** | **DF** | **MS** | **F** | **P** |
| Location | $S_A^2 = sm \sum\limits_{i=1}^{r} (\overline{X}_i - \overline{X})^2$ | $r-1$ | $\overline{S}_A^2 = \frac{S_A^2}{r-1}$ | $F_A = \overline{S}_A^2 / \overline{S}_E^2$ | $P_A = P(F_A > F_{1-p_A})$ |
| Error | $S_E^2 = \sum\limits_{i=1}^{r} \sum\limits_{j=1}^{s} \sum\limits_{k=1}^{m} (X_{ijk} - \overline{X}_{ij})^2$ | $rs(m-1)$ | $\overline{S}_E^2 = \frac{S_E^2}{rs(m-1)}$ | / | / |
| Time | $S_B^2 = rm \sum\limits_{j=1}^{r} (\overline{X}_j - \overline{X})^2$ | $s-1$ | $\overline{S}_B^2 = \frac{S_B^2}{s-1}$ | $F_B = \overline{S}_B^2 / \overline{S}_E^2$ | $P_B = P(F_B > F_{1-p_B})$ |
| Error | $S_E^2 = \sum\limits_{i=1}^{r} \sum\limits_{j=1}^{s} \sum\limits_{k=1}^{m} (X_{ijk} - \overline{X}_{ij})^2$ | $rs(m-1)$ | $\overline{S}_E^2 = \frac{S_E^2}{rs(m-1)}$ | / | / |

**Table 4.** One-way ANOVA of monthly and seasonal $CO_2$ concentration of urban residential green space.

| Quadratic Sum | Degree of Freedom | Mean Square | Mean Square Percentage | Probability ($\alpha$ = 0.05) |
|---|---|---|---|---|
| **SS** | **DF** | **MS** | **F** | **P** |
| $S_E^2 = \sum\limits_{i=1}^{r} n_i (\overline{Y}_i - \overline{Y})^2$ | $r-1$ | $\overline{S}_A^2 = \frac{S_A^2}{r-1}$ | $F = \frac{\overline{S}_A^2}{\overline{S}_E^2}$ | P |
| $S_E^2 = \sum\limits_{i=1}^{r} \sum\limits_{j=1}^{n_i} (\overline{Y}_{ij} - \overline{Y}_i)^2$ | $n-r$ | $\overline{S}_E^2 = \frac{S_E^2}{n-r}$ | / | / |
| $S_E^2 = \sum\limits_{i=1}^{r} n_i (Y_{ij} - \overline{Y})^2$ | $n-1$ | / | / | / |

Where $r$ is the level number of factor A, that is, the number of months and seasons; $n$ is the total number of samples, $\overline{X}$ denotes the mean of samples, $n_j$ denotes the number of repeated experiments under the level $A_j$, $X_i$ is the average $CO_2$ concentration under the level $A_j$ (month and season), and $X_{ij}$ is the $CO_2$ concentration in the $j$th region in the $i$th month or season

## 3. Results and Discussion

### 3.1. Diurnal and Gradient Variations of $CO_2$ Concentration in Urban Residential Green Space

(1) $CO_2$ concentration in urban residential green space during plant growing season.

The $CO_2$ concentration in the urban residential green space of the sampling points firstly exhibits a significant decreasing trend from 8:00 AM to 14:00 PM followed by a boom up until 18:00 PM (Figure 2). The maximum daily variation of the sample points and the comparison sample points are 118 $\mu mol \cdot mol^{-1}$ (occurred in XRO) and 68 $\mu mol \cdot mol^{-1}$,

respectively. Additionally, there is a significant difference in the $CO_2$ concentration during the changes of the measuring times and locations. It is also noted that among the four measured urban residential communities, the highest to the lowest measured $CO_2$ concentrations occurred in GBH, TMC, HNB, and XRO, respectively.

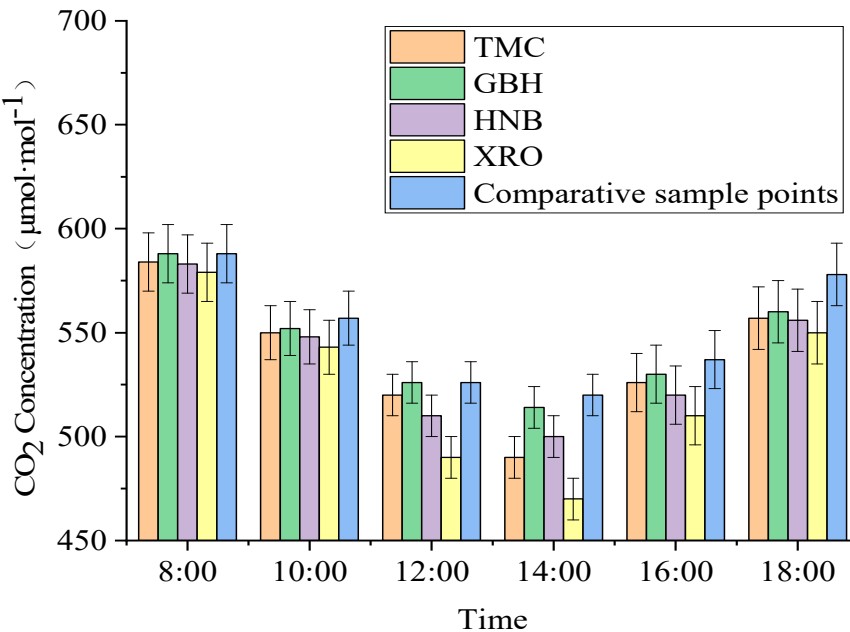

**Figure 2.** $CO_2$ concentration of sample points and comparison points in green space of urban residential area during growth season ($\mu mol \cdot mol^{-1}$).

In addition, the $CO_2$ concentration of sample points in the core zone for each urban residential community are all less than those in the transition zone and the edge zone (Figure 3). The variation trend of $CO_2$ concentration in the comparison sample points are similar to those in the four urban residential areas. However, it is note-worthy to stress that for the comparison sample points, the variation range is lower than that measured in the core area, which is 68 $\mu mol \cdot mol^{-1}$.

Specifically, in Figure 2, it is observed that the for all sampling points, the $CO_2$ concentration are gradually declined between 8:00 a.m. to 12:00 a.m. After the $CO_2$ concentration reached the lowest point at 14:00 PM, the values tended to increase slowly.

By comparing the $CO_2$ concentration in different locations during the plant growth season, it can be found that for all the locations, including the core, transition, and edge zones, the $CO_2$ concentrations are increasing significantly. Moreover, although both the measuring locations and the times can impact the $CO_2$ concentrations, different measuring times will cause greater changes in $CO_2$ concentration (Table 5).

**Table 5.** Regional samples and variance analysis of $CO_2$ concentration in the growth season of urban residential green space.

| | Quadratic Sum | Degree of Freedom | Mean Square | Mean Square Percentage | Probability ($\alpha$ = 0.05) |
|---|---|---|---|---|---|
| | **SS** | **DF** | **MS** | **F** | **P** |
| Location | 11,542.61 | 3 | 3847.54 | 16.15 | $5.76 \times 10^{-4}$ |
| Error | 2144.76 | 9 | 238.31 | / | / |
| Time | 79,906.97 | 5 | 15,981.39 | 143.39 | $4.11 \times 10^{-12}$ |
| Error | 1671.82 | 15 | 111.45 | / | / |

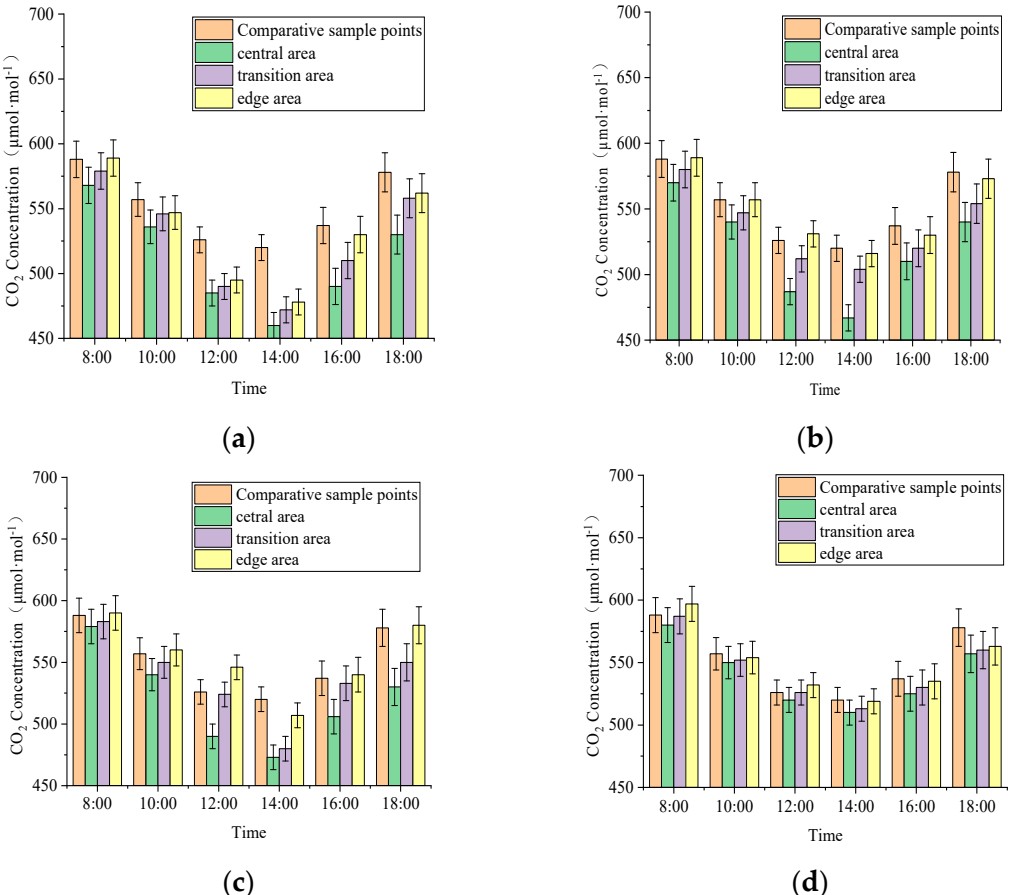

**Figure 3.** $CO_2$ concentration of regional samples and comparison samples in urban residential green space during the growth season ($\mu mol \cdot mol^{-1}$): (**a**) XRO; (**b**) HNB; (**c**) TMC; (**d**) GBH.

(2) $CO_2$ concentration in urban residential green space during plant non-growing season.

The overall change pattern shows a trend of decreasing first and subsequent increasing, which is similar to the corresponding change result in the plant growing season (Figure 4). In addition, the $CO_2$ concentration at all sample points are higher than those measured during the growing season.

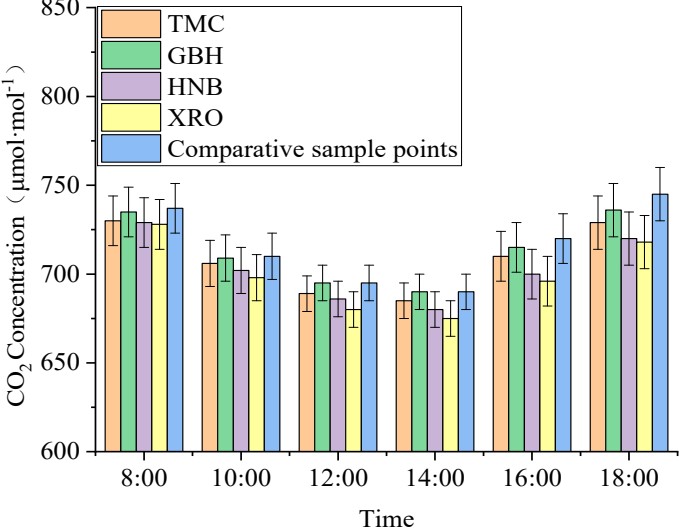

**Figure 4.** $CO_2$ concentration of sample points and comparison points in urban residential green space during non-growing season ($\mu mol \cdot mol^{-1}$).

According to Figure 5, it is demonstrated that the variating trend of the measured $CO_2$ concentration during 8:00–12:00 in the plant non-growing season is similar to that of growing season (Figure 5). In detail, the measured results for all the sampling points displayed a trend of firstly increasing before 14:00 PM and then declining. Compared with core and the transition zones, the largest decreases were found in the edge zone.

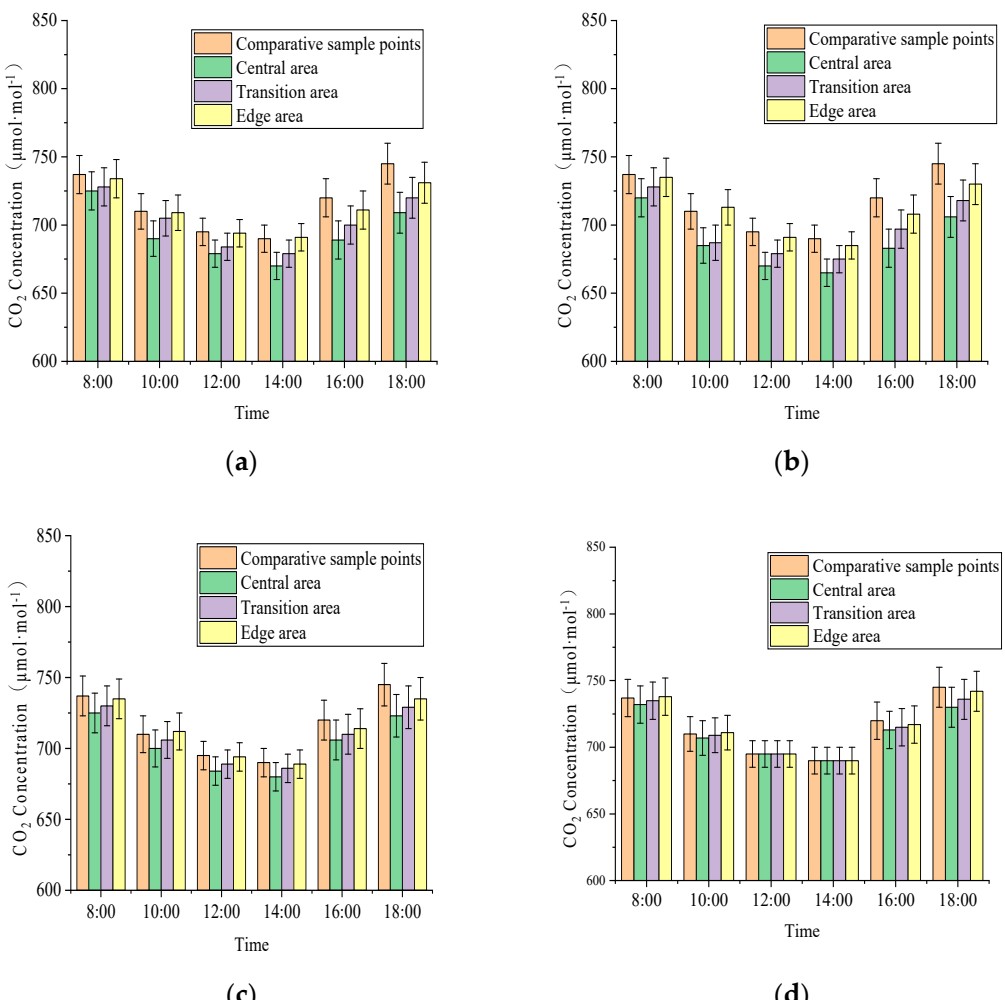

**Figure 5.** $CO_2$ concentration of regional samples and comparison samples in urban residential green space during non-growing season (μmol·mol$^{-1}$): (**a**)XRO; (**b**)HNB; (**c**)TMC; (**d**) GBH.

It is also worth stressing that when compared with the measurement location, the measuring time yields a more significant impact, and the overall influence is lower in the plant non-growing season than in the plant growing season (Table 6).

**Table 6.** Analysis of variance of $CO_2$ concentration in non-growing season regional samples and time of urban residential green space.

|  | Quadratic Sum | Degree of Freedom | Mean Square | Mean Square Percentage | Probability ($\alpha = 0.05$) |
|---|---|---|---|---|---|
|  | **SS** | **DF** | **MS** | **F** | **P** |
| Location | 4218.13 | 3 | 1406.04 | 11.10 | 0.002 |
| Error | 1140.46 | 9 | 126.72 | / | / |
| Time | 32,113.88 | 5 | 6422.78 | 521.12 | $2.94 \times 10^{-16}$ |
| Error | 184.88 | 15 | 12.33 | / | / |

### 3.2. Monthly Variation Characteristics of $CO_2$ Concentration for Green Space Plants in Residential Area

The monthly average $CO_2$ concentration and the relative analysis of variances from May 2019 to May 2020 are illustrated in Figure 6 and Table 7, respectively. From Figure 6, it is observed that significant diversities occurred in different measured locations. Specifically, the highest $CO_2$ concentrations were found in November, December, and January. In detail, in December, the average $CO_2$ concentration of GBH reached 718 μmol·mol$^{-1}$, while that obtained from a comparison sampling point is 720 μmol·mol$^{-1}$. The measurement results of $CO_2$ concentrations in May, June, July and August were comparatively low. For example, the average minimum $CO_2$ concentration of XRO in July was 400 μmol·mol$^{-1}$, and that of the comparison point was 490 μmol·mol$^{-1}$. Moreover, the monthly average $CO_2$ concentration measured from the sampling points were lower than the comparison sampling points. In Table 7, it was found that there were significant changes in $CO_2$ concentration gradients in terms of months with different measured locations (Table 7).

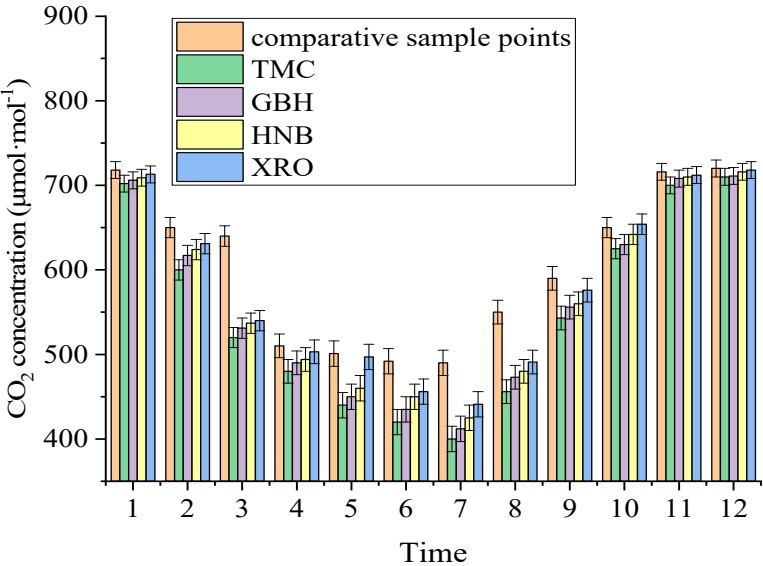

**Figure 6.** Monthly $CO_2$ concentration in residential area sample points (μmol·mol$^{-1}$).

**Table 7.** Variance analysis of monthly $CO_2$ concentration of regional sample points in urban residential green space.

|  | Quadratic Sum | Degree of Freedom | Mean Square | Mean Square Percentage | Probability ($\alpha = 0.05$) |
|---|---|---|---|---|---|
|  | **SS** | **DF** | **MS** | **F** | **P** |
| Inter-group | 614,779.65 | 11 | 55,889.06 | 89.88 | 0 |
| Intra-group | 29,848.00 | 48 | 621.83 | / | / |
| Sum | 644,627.65 | 59 | / | / | / |

### 3.3. Seasonal Variation Characteristics of $CO_2$ Concentration in Residential Green Space

It was found that the $CO_2$ concentration varied significantly in different seasons (Figure 7). In detail, the corresponding location and season with the lowest value were found to be XRO and summer, with an average value of 450 μmol·mol$^{-1}$, which is lower than the comparative sampling point, 501 μmol·mol$^{-1}$. Moreover, the $CO_2$ concentration for all sampling points only presented a small difference in winter, and the measured values of the sampling points were all smaller than those of the comparison sampling points. The results of seasonal variance analysis are listed in Table 8.

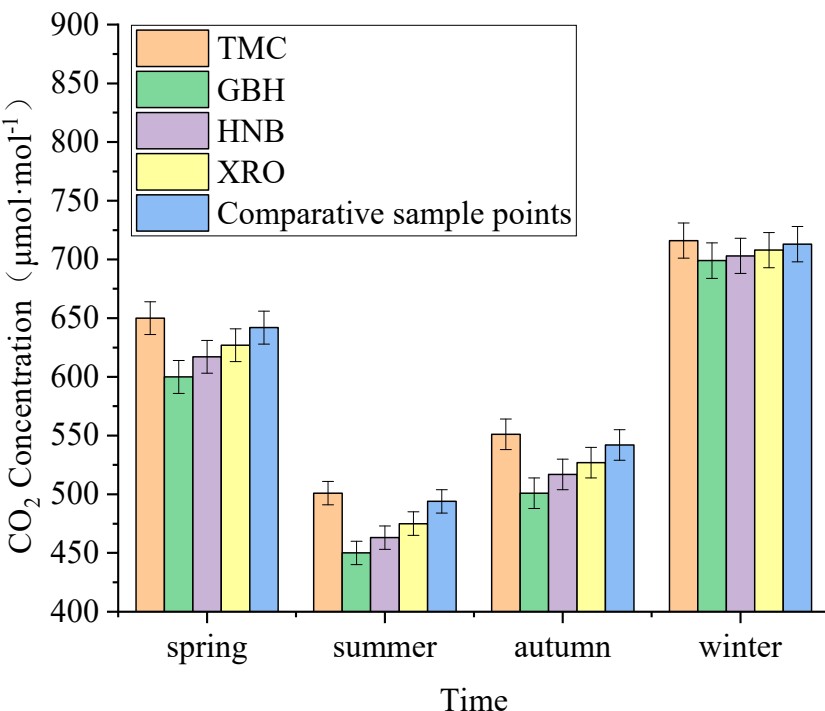

**Figure 7.** $CO_2$ concentration in different seasons in residential areas ($\mu mol \cdot mol^{-1}$).

**Table 8.** Variance analysis of $CO_2$ concentration in different seasons of regional samples in urban residential green space.

|  | Quadratic Sum | Degree of Freedom | Mean Square | Mean Square Percentage | Probability ($\alpha = 0.05$) |
|---|---|---|---|---|---|
|  | SS | DF | MS | F | P |
| Inter-group | 159,529.20 | 3 | 53176.40 | 165.34 | $2.99 \times 10^{-12}$ |
| Intra-group | 5146.00 | 16 | 321.63 | / | / |
| Sum | 164,675.20 | 19 | / | / | / |

### 3.4. $CO_2$ Concentration and Plant Community Structure of Urban Residential Green Space

As can be seen from Table 9, among all the measured sampling points, the lowest value of $CO_2$ concentration occurred in the plant community structure of trees-shrubs-grass with 692.25 $\mu mol \cdot mol^{-1}$, while the highest value emerged in the plant community structure of lawn with 720 $\mu mol \cdot mol^{-1}$. For the plant community structure of shrub-grass, the value of the $CO_2$ concentration was 702.5 $\mu mol \cdot mol^{-1}$. Notably, according to the variance analysis in Table 10, it was revealed that the $CO_2$ concentrations of the varied community structures experienced significant differences.

### 3.5. Analysis of the Relationship between Total Amount of Residential Green Space and $CO_2$ Concentration and Ecological Bearing Capacity

3.5.1. The Relationship between the Total Green Quantity of Residential Green Space and $CO_2$ Concentration

Residential green space is mainly composed of trees, shrubs, and grass. Considering the existence of trees in the ground space that mainly in the form of trunks, the ground green space is mainly composed of the shrubs and grass. In this research, the total green quantity is represented by the sum of grass ($A_{lawn}$), shrubs ($A_{shrub}$) and vertical projection area of the tree crowns ($A_{a1}$) in residential green space, which can be calculated by

$$A_{total \ green \ quantity} = A_{lawn} + A_{shrub} + A_{al} \qquad (1)$$

Considering the overlapping effects of above components, the minimum and maximum value of the greening volume ratio ($R_g$) are assumed to be 0 and 2 in this research, respectively, and it can be evaluated by the ratio between the summation of the areas of grass ($A_{lawn}$), shrubs ($A_{shrub}$), and the tree crowns ($A_{tree}$) on the total green quantity ($A_{green\ space}$), such as

$$R_g = (A_{lawn} + A_{shrub} + A_{tree})/A_{green\ space} \qquad (2)$$

**Table 9.** Plant community structure and $CO_2$ concentration in residential green space ($\mu mol \cdot mol^{-1}$).

| Types of Plant Community Structure | Sample Point | Concentration of $CO_2$ ($\mu mol \cdot mol^{-1}$) | |
|---|---|---|---|
| Comparison sample points 1 | / | 725 | 724 |
| Comparison sample points 2 | / | 723 | |
| Trees-shrubs-grass type | A | 685 | 692.25 |
| | D | 695 | |
| | F | 696 | |
| | G | 693 | |
| Shrubs-grass type | B | 700 | 702.5 |
| | C | 703 | |
| | E | 705 | |
| | H | 702 | |
| Lawn type | I | 713 | 716 |
| | J | 717 | |
| | K | 714 | |
| | L | 720 | |

**Table 10.** Variance analysis of $CO_2$ concentration of plant structure types of regional samples in urban residential green space.

| | Quadratic Sum | Degree of Freedom | Mean Square | Mean Square Percentage | Probability ($\alpha = 0.05$) |
|---|---|---|---|---|---|
| | SS | DF | MS | F | P |
| Inter-group | 1849.75 | 3 | 616.58 | 51.49 | $2.19 \times 10^{-6}$ |
| Intra-group | 119.75 | 10 | 11.98 | / | / |
| Sum | 1969.50 | 13 | / | / | / |

Table 11 shows the calculation results of different varieties of vegetation, plant areas, total green quantity, and greening volume ratio from the sampling points in urban residential green space:

**Table 11.** Plant areas, total green quantity, and green capacity in sample points area of residential green space.

| Residential Communities | Area of Total Green Space (m$^2$) | Trees (m$^2$) | Shrubs (m$^2$) | Grass (m$^2$) | Total Green Quantity (m$^2$) | Greening Volume Ratio |
|---|---|---|---|---|---|---|
| GBH | 5269 | 1580.7 | 2107.6 | 3161.4 | 6849.7 | 1.3 |
| TMC | 5589 | 2828.3 | 3543.1 | 2045.9 | 8417.3 | 1.5 |
| HNB | 6677 | 4036.7 | 3220 | 3457 | 10,713 | 1.6 |
| XRO | 6720 | 4060.2 | 3683.8 | 3036.2 | 10,780 | 1.6 |

Plants can significantly reduce the concentration of $CO_2$ in the ecological environment, which is imperative for the achievement of carbon and oxygen balance in cities. As can be seen from Figure 8, the greater the total green number of plants in residential areas, the higher the carbon sequestration amount and the lower the $CO_2$ concentration. In addition,

the total amount of trees is the main factor that reflected the total amount of green per unit area, which contributed more to the total amount of green than shrubs and grass. Moreover, in regard to the total green quantity, the green plot ratio reflects the quality of green space. The lower the $CO_2$ concentration, the larger the green area or the higher the green area ratio. In terms of carbon sink of urban residential green space, vegetation in residential green spaces is the main part of fixed and accumulated $CO_2$. The influencing factors of plant carbon sequestration capacity mainly include vegetation species, growth years, and community structures. In addition, environmental climate, external disturbance, vegetation coverage, and layout rationality can certainly affect the carbon sequestration ability of plants as well.

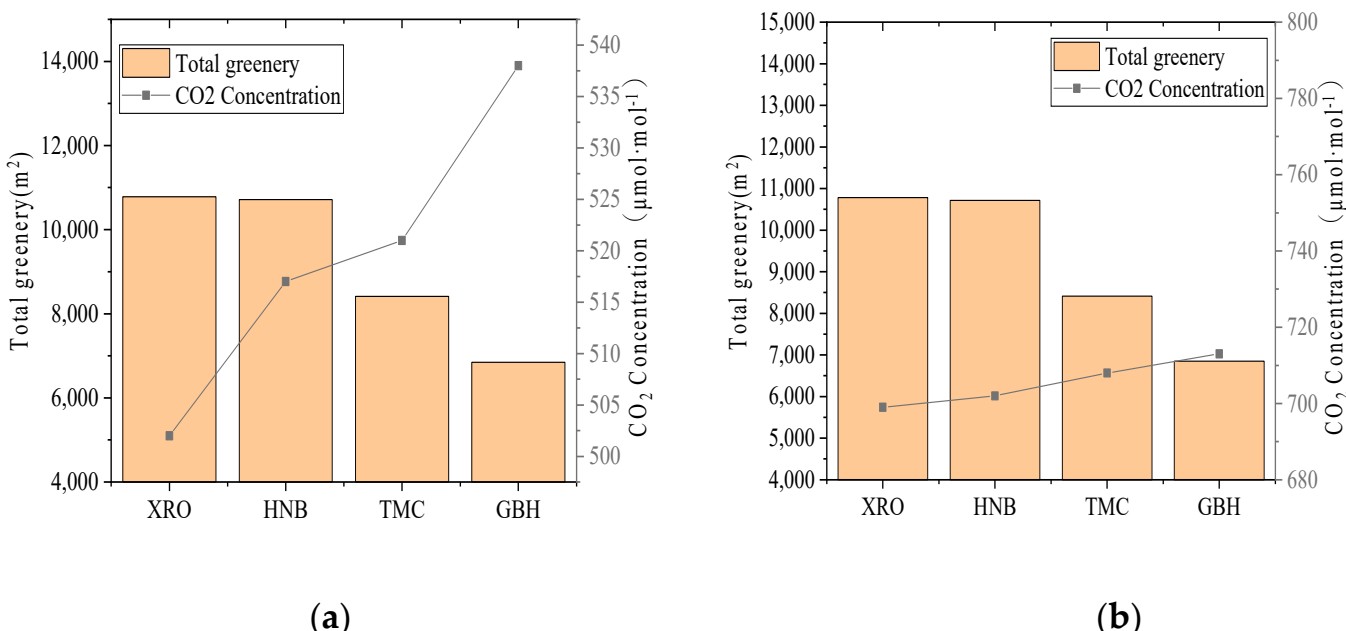

(**a**)  (**b**)

**Figure 8.** Relationship between daily average $CO_2$ concentration and total green quantity in plant growing season and non-growing season: (**a**) Plant growing season; (**b**) Plant non-growing season.

Specifically, it can also be observed in Figure 9 that the highest to lowest carbon sequestration capacities for different varieties of vegetation are trees > shrubs > grass.

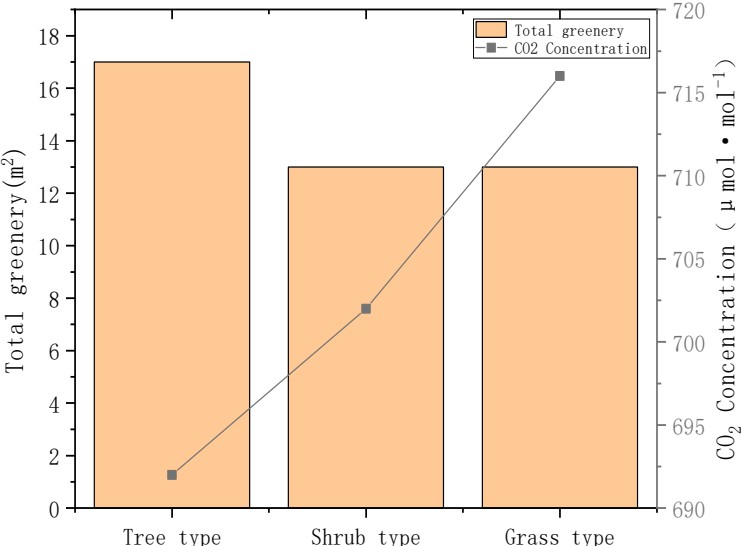

**Figure 9.** Relationship between $CO_2$ concentration of different plant types per 10 m$^2$ and total green quantity of residential green space.

3.5.2. Analysis of the Relationship between Total Green Quantity and Bearing Capacity of Residential Green Space

The classification of environmental bearing capacity of residential green space is shown in Table 12, and the calculation value can be divided into excessive overload, overload, full load, reasonable bearing capacity, and good bearing capacity.

**Table 12.** Classification standards of $CO_2$ bearing capacity of urban residential green space.

| Capacity Evaluation Value | Description |
| --- | --- |
| evaluation value > 3.0 | excessive overload |
| 1.0 < evaluation value < 3.0 | overload |
| evaluation value = 1 | full load |
| 0.7 < evaluation value < 1.0 | reasonable bearing |
| evaluation value < 0.7 | well bearing |

The actual bearing capacity of residential green space to $CO_2$ concentration is determined by the maximum number of pollutants it can accept. Small amounts of $CO_2$ will not pose a threat to human survival, but if human beings and other organisms inhale too much $CO_2$ in a short period of time, it will cause different level of risk. The relationship between $CO_2$ concentration and human comfort is shown in Table 13:

**Table 13.** $CO_2$ concentration versus human comfort.

| Degree | Human Discomfort Symptoms at Different $CO_2$ Concentrations | Symptom |
| --- | --- | --- |
| 1 | 350–400 µmol·mol$^{-1}$ | Healthy |
| 2 | 400–700 µmol·mol$^{-1}$ | Normal level |
| 3 | 700~1000 µmol·mol$^{-1}$ | Acceptable |
| 4 | 1000–200 µmol·mol$^{-1}$ | Tired and discomfort |
| 5 | 2000–4000 µmol·mol$^{-1}$ | Dyspneic |

As can be seen from the above table, the critical value of $CO_2$ concentration for human comfort is 700 ppm.

For the index of capacity $R_i$, it can be calculated as following equation and the calculated results are listed in Table 14

$$R_i = C_i / C_{i0} \tag{3}$$

in which the $R_i$ denotes the index of capacity (refers to relative occupancy ratio of $CO_2$ concentration on ecological environmental bearing capacity; $C_i$ is the measured result of the $CO_2$ concentration; $C_{i0}$ is the critical value of the $CO_2$ concentration, which can be directly obtained from Table 13; when $R_i$ is less than 1, the degree of ecological environmental bearing capacity is overloading.

**Table 14.** Index of bearing capacity of green space in each residential area.

|  | XRO | HNB | TMC | GBH |
|---|---|---|---|---|
| Index of bearing capacity | 0.78 | 0.8 | 0.81 | 0.82 |
| Degree of bearing | Reasonable bearing | Reasonable bearing | Reasonable bearing | Reasonable bearing |

Results in Table 14 reveals that the green space of the four residential areas studied in this paper are in a reasonable bearing state at present and fall short of the requirements of well bearing capacity. Therefore, the bearing capacity of the corresponding residential green space to $CO_2$ concentration can be improved by increasing a certain area of ecologically complex green spaces of trees-shrubs-grass.

## 4. Conclusions

In this paper, experimental investigations of $CO_2$ concentration in typical urban residential communities were reported. By taking into account the ecological carbon sequestration capacity of green spaces, its benefits and efficiencies were explored by dint of analyzing several dominant influence factors, including total green quantity, location, and plant community structure. The key conclusions can be drawn as follows:

(1) The $CO_2$ concentration in urban residential area is essentially dependent on the location, measuring time, and plant community structure of the green space. The total green quantity has a positive correlation with carbon sequestration capacity;

(2) The maximum and minimum $CO_2$ concentrations emerged in winter and summer, respectively. Such seasonal variations were attributed to human activities and plant phenology in urban residential communities;

(3) Plant community structures with trees-shrubs-grass had the highest carbon sequestration capacity compared with other types. In same ground area, ecologically complex green spaces containing shrubs, trees and grasses performed much better than simple green spaces only containing single plant species such as grass;

(4) The green spaces in urban residential areas studied in this paper were all in the degree of reasonable bearing. In order to improve the local ecological environment, planning authorities could plant ecologically complex green spaces instead of the current plant community structures.

**Author Contributions:** Data curation, Y.G.; Funding acquisition, X.L.; Writing—original draft, Y.G.; Writing—review & editing, X.L. All authors have read and agreed to the published version of the manuscript.

**Funding:** This study was funded by the National Natural Science Foundation of China (grant numbers 52178179, 51778632 and U1934217), China Postdoctoral Science Foundation (grant numbers 2017T100647 and 2018M642658).

**Conflicts of Interest:** The authors declare no conflict of interest.

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
