# Peer review of "Experimental Study on the Carbon Sequestration Benefit in Urban Residential Green Space Based on Urban Ecological Carrying Capacity"

_sustainability, doi:10.3390/su14137780_

Round 1
Reviewer 1 Report
The paper is presenting an interesting study on the carbon sequestration benefit of urban residential green space and its evaluation index based on urban ecological carrying capacity. It is a complete work and a timely research with extensive literature review and comprehensive analyses providing new contributions to the body of knowledge in the associated field.
Reviewer 2 Report
Based on different vegetation communities in human residential green space, the diurnal, monthly and seasonal changes of CO2 concentration were studied to reveal the main driving factors of carbon sequestration ability. In addition, according to the concept of urban ecological carrying capacity, an index reflecting the carbon sequestration demand of urban residential areas was proposed in this paper. This paper has important reference and guiding significance for the construction of vegetation community composition and sustainable development of ecological environment in residential areas, but it also needs to be revised significantly before acceptance to publication.See the document for more specific reviewer comments.

Reviewer 3 Report
In general, it is a interesting article; however, there are several parts that the author must review such as title, abstract, definition of vegetation types as well as specific comments that are indicated in the PDF file.

Round 2
Reviewer 2 Report
The revised version is fine.
